

# Identification and expression analysis of the sucrose synthase gene family in pomegranate (*Punica granatum* L.)

Longbo Liu and Jie Zheng

School of Life Science, Huaibei Normal University, Huaibei, Anhui, China

## ABSTRACT

**Background.** Sucrose synthase (SUS, EC 2.4.1.13) is one of the major enzymes of sucrose metabolism in higher plants. It has been associated with C allocation, biomass accumulation, and sink strength. The *SUS* gene families have been broadly explored and characterized in a number of plants. The pomegranate (*Punica granatum*) genome is known, however, it lacks a comprehensive study on its *SUS* genes family.

**Methods.** *PgSUS* genes were identified from the pomegranate genome using a genome-wide search method. The *PgSUS* gene family was comprehensively analyzed by physicochemical properties, evolutionary relationship, gene structure, conserved motifs and domains, protein structure, syntenic relationships, and *cis*-acting elements using bioinformatics methods. The expression pattern of the *PgSUS* gene in different organs and fruit development stages were assayed with RNA-seq obtained from the NCBI SRA database as well as real-time quantitative polymerase chain reaction (qPCR).

**Results.** Five pomegranate *SUS* genes, located on four different chromosomes, were divided into three subgroupsaccording to the classification of other seven species. The *PgSUS* family was found to be highly conserved during evolution after studying the gene structure, motifs, and domain analysis. Furthermore, the predicted PgSUS proteins showed similar secondary and tertiary structures. Syntenic analysis demonstrated that four *PgSUS* genes showed syntenic relationships with four species, with the exception of *PgSUS2*. Predictive promoter analysis indicated that *PgSUS* gene*s* may be responsive to light, hormone signaling, and stress stimulation. RNA-seq analysis revealed that *PgSUS1/3/4* were highly expressed in sink organs, including the root, flower, and fruit, and particularly in the outer seed coats. qPCR analysis showed also that *PgSUS1*, *PgSUS3*, and *PgSUS4* were remarkably expressed during fruit seed coat development. Our results provide a systematic overview of the *PgSUS* gene family in pomegranate, developing the framework for further research and use of functional *PgSUS* genes.

# INTRODUCTION

Sucrose is the most common form of carbohydrate produced by photosynthetic leaves. It is imported into non-photosynthetic organs (sink organs) through the phloem (*Lutfiyya et al., 2007*). Sucrose has been acknowledged as a valuable carbon and energy source for various metabolic pathways related to plant growth and development, such as cell division,

Corresponding author
Jie Zheng, Zhengj@chnu.edu.cn

vascular tissue differentiation, seed germination, flowering induction, fruit development, anthocyanin synthesis, storage products accumulation, biotic and abiotic stresses response, and damage recovery (*Wang et al., 2015*). Therefore, the study of sucrose metabolism is beneficial for understanding numerous aspects of plant physiology.

Sucrose synthase (SUS) and invertase (INV) are widely regarded as two key enzymes for the sucrose cleavage reaction. INV catalyzes the irreversible hydrolyzation of sucrose into glucose and fructose (*Hirose, Scofield & Terao, 2008*), whereas SUS catalyzes the reversible cleavage of sucrose using uridine diphosphate (UDP) to yield fructose and UDP-glucose (*Stein & Granot, 2019*). These enzymes are tightly linked with phloem sucrose unloading (*Wang et al., 2015*). SUS activity is highly associated with C allocation, biomass accumulation, and sink strength (*Stein & Granot, 2019*). For instance, the deletion or suppression of the *SUS* gene decreases maize seed weight (*Chourey et al., 1998*), reduces pea seed mass (*Craig et al., 1999*), leads to tomato fruit setting abnormality (*D'Aoust, Yelle & Nguyen-Quoc, 1999*), inhibits stem thickening in *Populus tomentosa* (*Li et al., 2020*), and reduces the stem height, diameter, and biomass in aspen (*Dominguez et al., 2021*). The overexpression of *SUS* increases the growth rate and facilitates plant biomass accumulation in *Arabidopsis* (*Xu & Joshi, 2010*), promotes cellulose biosynthesis and increases the lodging resistance in tobacco stem (*Wei et al., 2015*), and accelerates vegetative growth, thickens the secondary cell wall, and increases the stem breaking force in poplar (*Li et al., 2019*). SUS also plays important roles in sugar metabolism during fruit development. Citrus *CitSus1* and *CitSus2* (*Islam et al., 2014*), peach *PpSUS1*, *PpSUS3*, and *PpSUS5* (*Zhang et al., 2015*), pear *PbrSUS2* and *PbrSUS15* (*Lv et al., 2018*), and apple *MdSUS1s* and *MdSUS2.1* (*Tong et al., 2018*) are all thought to be responsible for the sucrose download and partitioning in fruits. Strawberry fruits with the suppression of *FaSUS1* showed significantly delayed fruit ripening, and downregulated sucrose and anthocyanin contents (*Zhao et al., 2017*). Additionally, the SUS enzyme is thought to participate in the regulation of several important metabolic processes, such as cellulose and callose synthase, nitrogen fixation, abiotic stresses response, and development of shoot apical meristem (*Stein & Granot, 2019*).

Sucrose synthase is encoded by a small, multigene family in both monocot and dicot species. The number of *SUS* gene family members to date differs among the plant species. In maize, only three *SUS* genes have been identified (*Duncan, Hardin & Huber, 2006*), however, five *SUS* genes have been found in grape (*Zhu et al., 2017*). *Arabidopsis*, rice, cacao, peach, tomato, and citrus all contain a *SUS* genes family with six *SUS* genes (*Baud, Vaultier & Rochat, 2004*; *Hirose, Scofield & Terao, 2008*; *Li et al., 2015*; *Zhang et al., 2015*; *Goren et al., 2017*; *Duan et al., 2021*; *Islam et al., 2014*), whereas seven, 11, 14, and 15 *SUS* genes were found in cotton (*Chen et al., 2012*), apple (*Tong et al., 2018*), Indian mustard (*Koramutla et al., 2019*), and poplar (*An et al., 2014*), respectively. In all cases, *SUS* genes showed structural conservation but functional divergence during evolution according to the physical and chemical properties of gene and protein structures, phylogenetic relatedness, and spatial–temporal expression patterns (*Xu et al., 2019*). The *SUS* gene family has been extensively studied in various plants. However, the *SUS* genes in pomegranate not yet been described.

Pomegranate (*Punica granatum* L.) is an ancient perennial plant species of the *Punicaceae* family that has become an emerging edible fruit crop due to its good environmental adaptation and wide medicinal applications (*Conidi, Drioli & Cassano, 2020*). The global pomegranate market is promising, with an expected 14% annual growth rate, and is expected to reach 23.14 billion United States dollars (USD) by year 2026 (*Conidi, Drioli & Cassano, 2020*). Improving the fruit quality is important to enhance the market competitiveness of pomegranate production. Particularly, the accumulation of sugar content is key in determining the taste, flavor, and value for most fleshy fruit crops (*Li, Feng & Cheng, 2012*). Therefore, the comprehensive analysis of sucrose synthase genes may improve the understanding of its molecular function and identify the key genes involved in pomegranate fruit sugar metabolism. Recently, the high-quality genome data of several cultivars of pomegranate have been released, including those of 'Dabenzi', 'Taishanhong', and 'Tunisia', which supplies genome data for further the molecular function identification of pomegranate genes (*Qin et al., 2017*; *Yuan et al., 2018*; *Luo et al., 2020*). Here, we identified and characterized five *SUS* genes on the pomegranate genome-wide scale and investigated their expression patterns. This study focused on *PgSUS* member isolation and identification, evolutionary relationships, exon/intron arrangement, conserved motif and domain, protein structure, synteny relationship, promoter elements, and expression patterns of the pomegranate *SUS* gene family. These results will provide insight for further investigations of the possible functions of the *SUS* gene family in pomegranate for regulating plant growth, particularly in the development and maturation of the fruit.

## MATERIALS & METHODS

### Obtaining genome sequences and identifying *PgSUS* family members in pomegranate

The genome sequences and annotation data of pomegranate cv. Tunisia were obtained from the NCBI genome database (https://www.ncbi.nlm.nih.gov/genome/13946?genome_assembly_id=720008) (*Luo et al., 2020*). Six known AtSUS proteins sequences were downloaded from TAIR database (http://www.arabidopsis.org/) and were used as a query to search against the pomegranate protein database (e-value $<1 \times 10^{-5}$, identify >50%). The search used a local BLAST alignment in order to identify potential members of *SUS* gene family in pomegranate. The hidden Markov model (HMM) profiles of the sucrose synthase domain (PF00862) and glycosyl transferases domain (PF00534) collected from the Pfam website (http://pfam.xfam.org/) were used as queries to search the candidate *PgSUS* from pomegranate proteins using HMMER 3.1 (e-value $< 1 \times 10^{-5}$) (*Finn et al., 2015*). The sucrose phosphate synthase (SPS) gene family with a sucrose-phosphatase domain (PF05116) in the N-terminal was also found to contain SUS protein conserved domains (PF00862 and PF00534). The resulting putative proteins were further examined by using the SMART and NCBI conserved domain database (CDD) (*Letunic & Bork, 2018*; *Lu et al., 2020*). We filtered out the candidates with a sucrose-phosphatase domain and those that lacked the sucrose synthase and glycosyl transferases domains.

The information on the pomegranate *SUS* chromosomal positions was obtained from the genome annotation data. The ExPasy website (http://web.expasy.org/protparam/) was used to evaluate the molecular weight (MW), isoelectric point (pI), instability index, aliphatic index, and grand average of hydropathicity (GRAVY). The NetPhos 3.1 server (http://www.cbs.dtu.dk/services/NetPhos/) was used to predicted the PgSUS proteins phosphorylation sites (*Blom et al., 2004*).

## Nucleotide and amino acid sequences alignment of *SUS* genes from eight species

The nucleotide and proteins sequences of 68 *SUS* genes were collected from *Arabidopsis thaliana* (6), *Oryza sativa* (6), *Glycine max* (12), *Malus domestica* (11), *Pyrus bretschneideri* (17), *Prunus persica* (6), *Vitis vinifera* (5), and *Punica granatum* (5), respectively. Multiple *SUS* genes sequence alignments were performed using the CLUSTAL_X program (http://www.clustal.org/).

## Phylogenetic analysis and classification of SUS gene family

A phylogenetic tree of 68 SUS proteins from eight species was generated using MEGA X software (http://www.megasoftware.net/). The tree was based on the maximum-likelihood (ML) method with the substitution model JTT+G+I and 1,000 bootstrap replications. PgSUS proteins were further categorized into different subfamilies according to the classification records of subfamily members of other species. The proteins sequences used in the phylogenetic analysis are listed in Data S1.

## Gene structure construction, conserved motif, domain, and protein structure analysis

The information on gene structure for each of the 68 *SUS* genes was extracted from their GFF3 files. This data included sequence length, number, and arrangement of exons and introns. The conserved motif type and sequence of the *SUS* family were analyzed by MEME (http://meme-suite.org/tools/meme). The phmmer protein database (https://www.ebi.ac.uk/Tools/hmmer/search/phmmer) was used to annotate the MEME motifs. The conserved domains of the SUS proteins were determined using SMART (http://smart.embl-heidelberg.de/). The gene structure, MEME, and conserved domain results were plotted with TBtools (*Chen et al., 2020*). Secondary and tertiary structures of PgSUS proteins were predicted using NPS@: SOPMA (https://www.predictprotein.org/signin) and the ExPaSy Swiss-Model online software (http://swissmodel.expasy.org), respectively.

## Syntenic analysis with four other species

MCScanX was used to obtain the syntenic relationships of five species: *Arabidopsis thaliana*, *Malus domestica*, *Pyrus bretschneideri*, *Vitis vinifera*, and *Punica granatum* (*Wang et al., 2012*). The results were presented with TBtools (*Chen et al., 2020*).

## *Cis*-acting element analysis of *PgSUS* genes promoter regions

We extracted 2,000 bp gene sequences of genomic DNA sequences upstream of the initiation codon (ATG). These were used to predict the putative *cis*-acting elements using PlantCARE online software (http://bioinformatics.psb.ugent.be/webtools/plantcare/html/).

## Expression pattern analysis of candidate *PgSUS* genes in pomegranate

Two published sets of transcriptome data were used to investigate the expression characteristics of the *PgSUS* genes. The abundance of the *PgSUS* transcripts of 12 samples, including root, leaf, flower, and three different development stages of the pericarp, inner, and outer seed coats (50, 95, and 140 days after flowering, DAF), were collected from the NCBI Sequence Read Archive database (accession number SRP100581) (*Qin et al., 2017*). The expression profiles of the *PgSUS* genes were analyzed at different developmental stages of the seed coats in pomegranate cultivars 'Dabenzi' and 'Tunisia'. These were collected at 50, 95, and 140 DAF and three biological replicates were collected per sample for RNA sequencing (accession number SRP212814, *Qin et al., 2020*). Clean reads of each sample were aligned to the pomegranate reference genome by HISAT2, using default parameter settings (*Kim et al., 2019*) after conducting a quality assessment of the filtered reads using Trimmomatic (*Bolger, Lohse & Usadel, 2014*). The mapped reads assembly of each sample was completed using StringTie (*Pertea et al., 2015*). The different gene expression levels were calculated according to transcripts per kilobase of exon model per million mapped reads (TPM). The TPM value was transformed into $\log_2$ (TPM + 1). The heatmap of the *PgSUS* genes expression was plotted using TBtools (*Chen et al., 2020*).

## Plant material

Samples were collected from three-year-old 'Tunisia' pomegranate trees at 26 °C under long-day conditions (14-h light/10-h dark) at approximately 60–70% humidity conditions. The trees were cultivated at the horticultural experimental station of Huaibei Normal University. We collected young root, mature leaves, and flowers. Healthy, uniform fruits were randomly collected at 45, 75, 115, and 150 DAF, respectively. Three replicates were prepared for each stage and each replicate contained 15 fruits. The fruit pericarp and seed coat were separated by hand. All samples were collected and immediately frozen in liquid nitrogen and stored at −80 °C.

## Total RNA isolation and quantitative PCR expression assay

Approximately 1 μg of high quality RNA per sample was extracted using plant RNA extraction kits (TIANGEN, China). The first strand of cDNA synthesis was performed using the TIANScript II RT kit (TIANGEN, China). We diluted 20 μL of cDNA from each sample to a total volume of 200 μL using DEPC water. These were used as qPCR templates. The reaction mixture contained1 μL cDNA, 0.5 μL each of the forward- and reverse-specific primer, 10 μL chamQ SYBR qPCR Master Mix (Vazyme, China), and 8 μL DEPC water, for a total volume 20 μL. The qPCR reaction was conducted in an ABI 7300 Real-Time PCR system with the following amplification program: 95 °C for 5 min, 40 cycles of 95 °C for 5 s, and 60 °C for 35 s. The pomegranate *PgActin* gene served as the reference gene, and the relative expressions levels of the genes were calculated according to *Livak & Schmitten (2001)*. Each sample was quantified in triplicate. Data were analyzed using SPSS software (22.0, USA) and Excel. All primers used for qPCR assay are shown in Data S2.

## RESULTS

### Identification of *PgSUS* genes

Two searches were performed to identify all possible *SUS* family members in the pomegranate genome. We obtained 14 putative *PgSUS* candidates by local Blast alignment according to query sequences of six *Arabidopsis* SUS proteins. Then, 19 PgSUS candidates were scanned from the pomegranate genome database based on the HMMER search. These two methods identified a total of 14 *PgSUS* candidates without a sucrose-phosphatase domain, which were verified with SMART and NCBI CDD databases. We found that 14 *PgSUS* candidates belonged to five genes and determined that each gene two-to-four transcripts after extraction and comparing the generic feature formats of these candidates. Finally, the five longest transcripts were isolated as the representative genes and were named *PgSUS1* to *PgSUS5*, according to their chromosomal information (Table 1).

Five *PgSUS* were dispersed on four chromosomes (Table 1). Among them, one single *SUS* gene originated from Chr2, 4, and 6 respectively, while the rest two were located in Chr8. cDNA length analysis of five *PgSUS* genes revealed variations from 4,249 bp (*PgSUS2*) to 7,426 bp (*PgSUS3*). However, the coding DNA sequence (CDS) lengths were similar, and ranged from approximately 2,418 bp (*PgSUS4*) to 2,706 bp (*PgSUS5*). Their proteins were composed of 805-901 amino acids, the putative molecular weights (MW) ranged from 92.26 kDa to 102.58 kDa, and the theoretical isoelectric points (pI) were approximately 5.99 to 8.19. The instability index of the five PgSUS proteins ranged between 32.35 and 42.23. The aliphatic index (A.i) were between 81.60 and 92.87 and all of the PgSUS proteins were hydrophilic (Table 1). Our prediction of the phosphorylation sites in PgSUSs showed that serine was the most common site for phosphorylation Tow typically serine phosphorylation sites were observed in all PgSUS proteins (Data S3).

The ClustalW2 program was used to align the nucleotide/amino acid sequences of five pomegranate SUS and 63 SUS members with seven other species. The comparison results showed that these 68 genes shared a high sequence homology at the nucleotide level (average 65.93% identity) as well as the protein level (average 65.16% identity) (Data S4). Among the five *PgSUS* genes, the nucleotide and amino acid sequences of *PgSUS1* were more similar to *PgSUS4* and their identity scores reached 80.98% and 89.94%, respectively. *PgSUS1* also showed similarity with *PgSUS3* with the sequence comparison scores of nucleotide and amino acid sequences of 68.03% and 69.44%, respectively. A pair of *PgSUS* genes (*PgSUS2-PgSUS5*) were also observed to be closely related (68.51% and 70.96% respectively) (Data S4).

### Phylogenetic analysis of *SUS* family members

We used five PgSUS from pomegranate, six AtSUS from *Arabidopsis*, and 57 other SUS proteins to construct the phylogenetic tree in order to clarify the evolutionary relationships. A total of 68 SUS proteins results from phylogenetic analysis were classified into three distinct subgroups categorized as SUS I, SUS II, and SUS III (Fig. 1). Corresponding to the nucleotide/amino acid sequence identity (Data S4), PgSUS1 was clustered with PgSUS4 to form the SUS I branch, which contained well-characterized *SUS* genes including *AtSUS1/2*, *PpSUS1/2/15*, and *VvSS4*. *PgSUS3* belonged to SUS II, which included *MdSUS2.1* and

Liu and Zheng (2022), *PeerJ*, DOI 10.7717/peerj.12814

**Table 1   The characteristics of the sucrose synthase genes in pomegranate.**

| Gene name | Gene ID | Genome location | cDNA length (bp) | CDS length (bp) | Protein length (aa) | MW (KDa) | Theoretical pI index | Instability index | Ai | GRAVY |
|---|---|---|---|---|---|---|---|---|---|---|
| PgSUS1 | XM_031527544.1 | Chr02:16303127…16309696 (-) | 6570 | 2421 | 806 | 92.78 | 6.56 | 33.79 | 92.25 | −0.262 |
| PgSUS2 | XM_031535599.1 | Chr04:6084363…6088611 (+) | 4249 | 2499 | 832 | 94.97 | 5.99 | 36.58 | 81.60 | −0.397 |
| PgSUS3 | XM_031544401.1 | Chr06:17237912…17245337 (-) | 7426 | 2433 | 810 | 92.26 | 5.99 | 42.23 | 89.10 | −0.249 |
| PgSUS4 | XM_031516902.1 | Chr08:5567456…5574103 (-) | 6648 | 2418 | 805 | 92.54 | 6.09 | 32.35 | 92.87 | −0.278 |
| PgSUS5 | XM_031551757.1 | Chr08:24807424…24812192 (-) | 4769 | 2706 | 901 | 102.58 | 8.19 | 39.61 | 83.77 | −0.348 |

Notes.

CDS, Coding DNA sequence; MW, molecular weight; pI, isoelectric point; Ai, aliphatic index; GRAVY, grand average of hydropathicity.

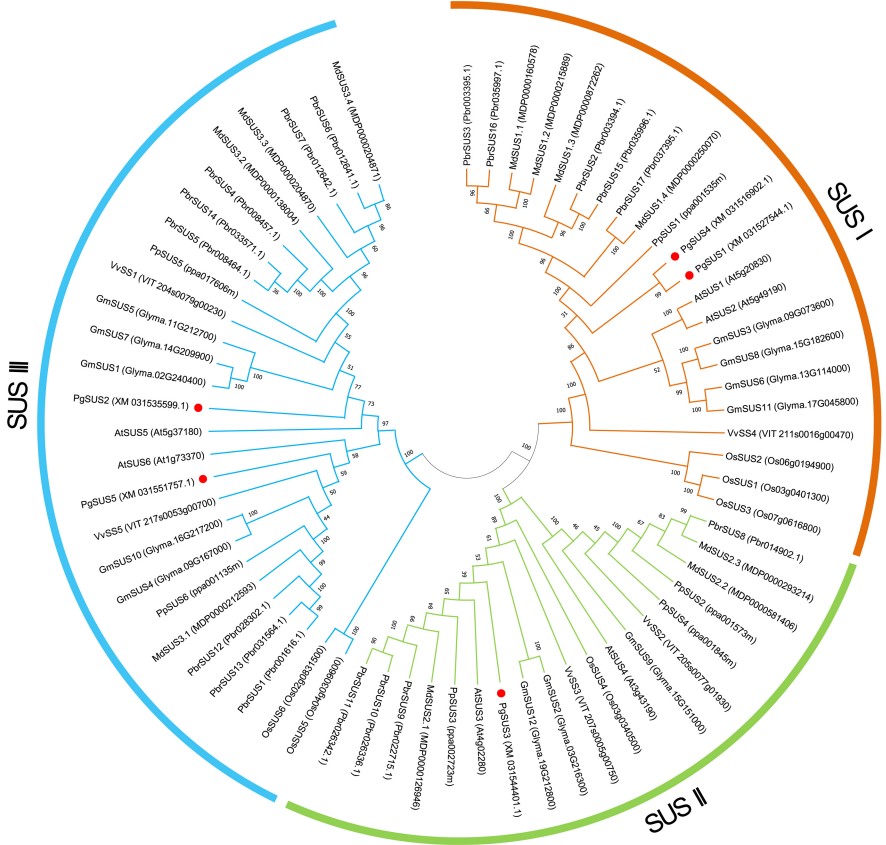

**Figure 1** **Phylogenetic relationship analysis of SUSs from pomegranate and seven other species.** The phylogenetic relationship was analyzed by MEGA X program based on the ML method JTT+G+I and 1,000 bootstrap replications. The block lines and orange, green, and blue arcs indicate the members in subgroups SUS I, SUS II, and SUS III, respectively. *PgSUS1* to *PgSUS5* are highlighted in red dots. The species names are abbreviations as follows: At, *Arabidopsis thaliana*; Os, *Oryza sativa*; Gm, *Glycine max*; Md, *Malus domestica*; Pbr, *Pyrus bretschneideri*; Pg, *Punica granatum*; Vv, *Vitis vinifera*, and Pp, *Prunus persica*.

*VvSS3*. Compared with the SUS I and SUS II subgroups, the genes clustered in the SUS III subgroup typically contained the proteins with a C-terminal extension, such as *PgSUS2/5*, *AtSUS5/6* and *MdSUS3.1/3.2/3.3* (Data S5). The results showed that although these *SUS* family genes shared high sequence similarities, including five pomegranate *SUS* genes, diversification was identified in this family through phylogenetic analysis.

## Gene structure, conserved motif, and domain analysis of *SUS* family genes

We further investigated the exons/introns exon/intron structure of all *SUS* genes to better understand the molecular evolution mechanism. These included five in pomegranate and 63 in other seven species according to the gene annotation files (Fig. 2A). *SUS* family gene sequences were split into approximately 15 exons in SUS I, and 14 exons in SUS II and SUS III genes, respectively, after taking introns loss into account (Fig. 2A; Data S6) (*Xu*

*et al., 2019*). The nucleotide sequences of 68 *SUS* genes showed high similarity (Data S4), therefore, high conservation of these gene structures was expected. Exons with lengths of 152/155, 193, 177/174/129, 117, 167 and 225, were highly conserved and arranged in same order in the CDS regions of all three *SUS* subgroups (Data S6). For each *SUS* subgroup, the gene structure also showed unique features: compared with SUS II genes, intron loss was a common phenomenon in SUS I and SUS III genes (Data S6). In the SUS I subgroup, exons with lengths of 336, 432, and 564, were split into two (119 and 217), three (119, 217 and 96) and two (322 and 245) exons in the SUS II subgroup, respectively. In the SUS III subgroup, exons with lengths of 567, were split into two exons (322 and 245) in the SUS II subgroup (Data S6). The exon sizes and splitting varied among the 3′ end of the genes of the SUS III subgroup. This was associated with the 3′ extension of SUS III proteins (Data S5; Data S6). In the *SUS* genes of pomegranate, the exons with lengths of 336 (or spilt into 119 and 217), 96, and 139 were typically conserved in *PgSUSs* (Data S4; Data S7). Moreover, in the same group, *PgSUS* genes showed a similar exon number, arrangement, and length with *SUS* genes from *Arabidopsis* (*Baud, Vaultier & Rochat, 2004*), apple (*Tong et al., 2018*), grape (*Zhu et al., 2017*), peach (*Zhang et al., 2015*), pear (*Lv et al., 2018*), soybean (*Xu et al., 2019*), and rice (*Hirose, Scofield & Terao, 2008*) (Fig. 2A; Data S6; Data S7).

We used the MEME online server to predict 15 motifs in the *SUS* gene family (Fig. 2B). Detailed information of these motifs is shown in Data S8. Among these motifs, the motifs 1, 3, 5, 6, 9, 10, 11, 12, and 13 represented the sucrose synthase domain; motifs 2 and 7 corresponded to the glycosyl-transferase domain, and the motif feature of motifs 4, 8, and 15 were unknown (Data S8). The majority of the SUS proteins from eight species contained the 14 predicted motifs, except motif 14, and showed a consistent array (Fig. 2B). Motifs 2 and 7, as the elements of the glycosyl-transferase domain signature, were highly conserved, suggested that these motifs are essential for enzyme function of sucrose synthase. However, several motifs which corresponded to the sucrose synthase domain were missing and the motif composition of some members were found to be variants in apple, pear, peach, and soybean (Fig. 2B). The five *PgSUS* members also shared common conserved motif compositions and had consistent arrangement (motifs 12, 9, 10, 11, 6, 3, 13, 1, 5, 7, 4, 2, 8 and 15) (Fig. 2B).

Tow typically conserved domains corresponding to the motifs features (sucrose synthase domain and the glycosyl-transferase domain) were screened in each member of 68 SUS proteins by matching with SMART and NCBI CDD (Fig. 2C). These two conserved domains were located at the N and C-terminal ends, respectively. This was consistent with the motif arrangement (Figs. 2B, 2C). In pomegranate, the length and distribution of two conserved domains of five SUS protein showed high consistency and conservation (Fig. 2C), indicating that they are critical for the function of PgSUS proteins.

## Prediction of protein structure of pomegranate SUS proteins

The secondary structures analysis showed that five pomegranate SUS proteins were composed of $\alpha$-helices, extended $\beta$ strands, $\beta$-turns, and random coils (Table 2; Data S9). The $\alpha$-helix was the major secondary structures among the five PgSUS proteins, accounting for 49.72–53.97%, followed by random coils (25.73–32.74%) and extended $\beta$ strands

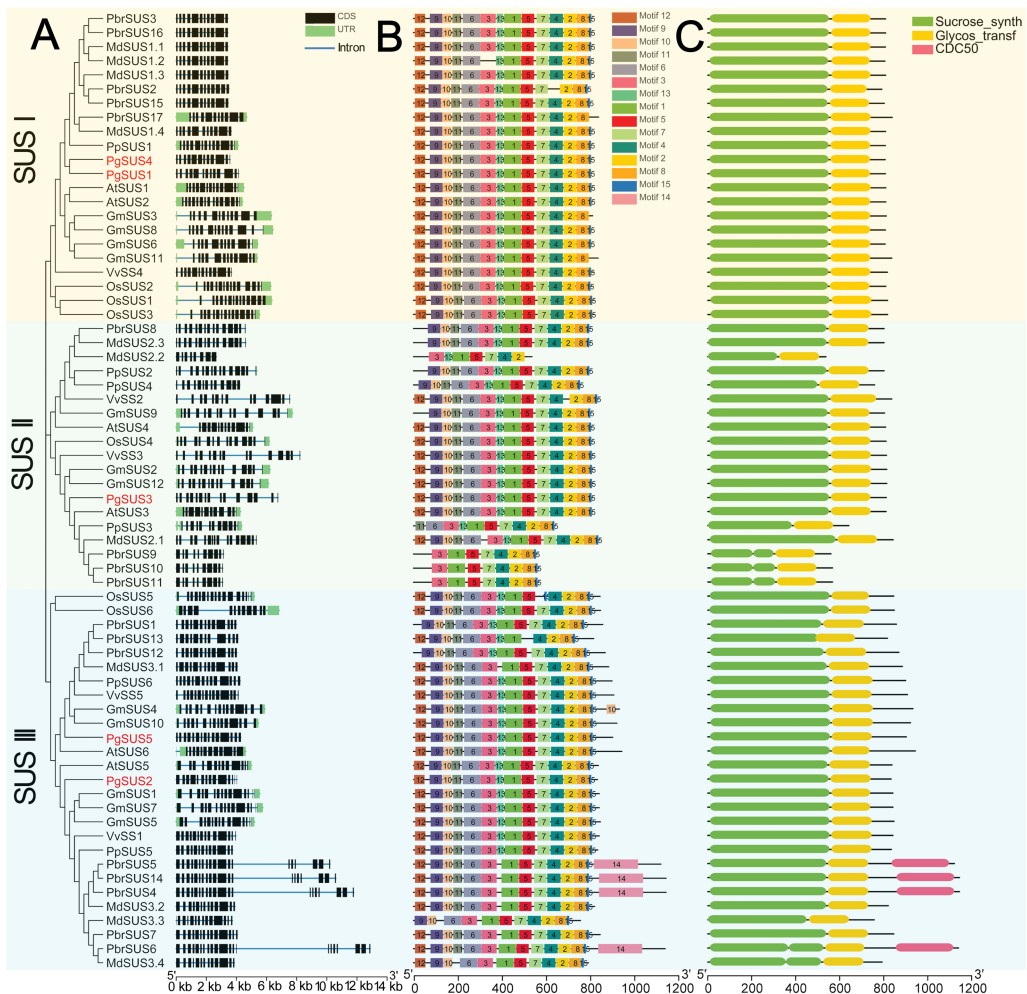

**Figure 2  Analysis of gene structure, conserved motif, and domain of *SUS* genes family in seven species.** (A) Exon/intron genomic structure of *SUS* genes. Exons, introns and untranslated regions (UTRs) are indicated by black rectangles, blue thin lines and green rectangles, respectively. (B) Composition and arrangement of the conserved motifs of SUS protein. Different colors and the numbers of the rectangles represent different motifs in the corresponding position of each SUS proteins. (C) Conserved domain structures of the SUS protein. The full-length protein sequences are indicated by thin black lines. The gene names *PgSUS1* to *PgSUS5* are highlighted in red.

(12.26–13.21%) (Table 2). These secondary structure distributions were also highly conserved in five PgSUS polypeptide chains (Data S9).

We predicted tertiary structures of the five pomegranate SUS proteins using the Swiss-model online software. The three-dimensional models of the PgSUSs proteins were based on templates 3s27 (Sucrose synthase) and 4rbn (Glycosyl transferases group 1). The results showed that the tertiary structure for PgSUS1 to PgSUS5 had two symmetric tetramers and comprised four main polypeptide chains. These were similar to PpSus1 to PpSus4 in peach (Data S10; *Zhang et al., 2015*). The 3D structure of PgSUS1 was quite similar with PgSUS4 among PgSUS1 to PgSUS5 (Data S10).

**Table 2   Secondary structural statistics of the PgSUS proteins.**

| Protein | Alpha helix (%) | Extended Beta strand (%) | Beta turn (%) | Random coil (%) |
|---------|-----------------|--------------------------|---------------|-----------------|
| PgSUS1  | 53.97           | 12.78                    | 7.82          | 25.73           |
| PgSUS2  | 52.76           | 12.26                    | 6.13          | 28.85           |
| PgSUS3  | 52.96           | 13.21                    | 6.67          | 27.16           |
| PgSUS4  | 53.42           | 13.04                    | 6.83          | 26.71           |
| PgSUS5  | 49.72           | 12.32                    | 5.22          | 32.74           |

## Syntenic analysis of five species *SUS* genes

We analyzed the syntenic relationships between pomegranate and four other species, including *A. thaliana*, *M. domestica*, *P. bretschneideri*, and *V. vinifera* to explore the evolutionary process of pomegranate *SUS* genes. Four *SUS* genes were found to have ten orthologous syntenic gene pairs in another four species (Fig. 3). *PgSUS1* was found to be syntenic with four genes from apple (*MdSUS1.1* and *MdSUS1.4*), pear (*PbrSUS17*), and grape (*VvSS4*). Three genes (*AtSUS6*, *MdSUS3.1* and *PbrSUS12*) showed synteny with *PgSUS5*, two genes (*PbrSUS17* and *VvSS4*) were syntenic with *PgSUS4*, and *PgSUS3* was syntenic only with *VvSS3* (Fig. 3). The syntenic relationships of these *SUS* orthologous gene pairs were consistent with their phylogenetic relationship (Fig. 1). However, *PgSUS2* in pomegranate was found to have no syntenic counterpart in the other four species. These results help to better understanding the possible roles of *SUS* gene family members in pomegranate.

## *Cis*-acting element analysis of *PgSUS* genes promoters

The *cis*-acting elements are crucial in the spatial–temporal and tissue-specific expression of genes. The *cis*-acting elements of the *PgSUS* genes were classified into five categories using the PlantCARE database The categories were: hormone responsive elements (HRE), tissue specific elements (TSE), light responsive elements (LRE), stress responsive elements (SRE), and others responsive elements (ORE) (Fig. 4). Detailed information of *cis*-acting elements in five *PgSUS* promoter regions is provided in Data S11. The number of LREs was the largest group (49%), followed by HREs (24%), SREs (15%), OREs (7%) and TSEs (6%) (Fig. 4A; Data S11). Among these, the presence of LREs was universal in five *PgSUS* genes promoters. The *PgSUS3* promoter contained 22 LREs, which was almost two times that of other *PgSUS* genes These results imply that *PgSUS3* may respond to light induction. Other *PgSUS* genes promoters contained several HREs, with the exception of *PgSUS2*. These hormones include abscisic acid (ABA), auxin, gibberellin (GA), methyl jasmonate (MeJA), and salicylic acid (SA). MeJA and ABA responsive elements were prevalent in the promoter regions of those four genes. Moreover, each *PgSUS* promoter contained tow-to-eight SREs, and were responsive to stresses including anoxic environments, low-temperatures, and drought (Fig. 4B; Data S11). In addition, *PgSUS* genes promoters also contained several OREs, such as circadian control, cell cycle regulation, and MYB binding sites, implying that *PgSUS* family genes may play multiple roles in plant growth and development.

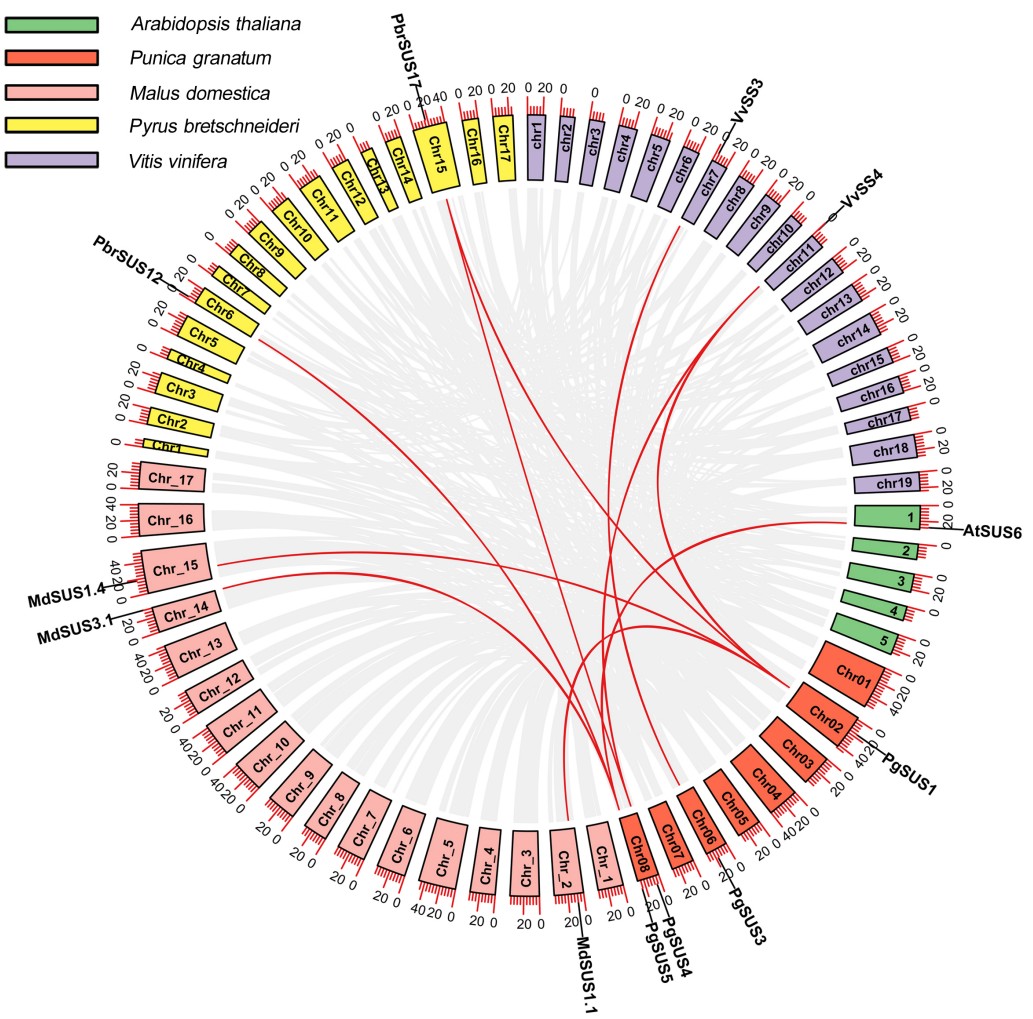

**Figure 3 Synteny analysis of *PgSUSs* genes with other four species.** The chromosomes of five species are marked with different colors. The short black lines on the circles represent the approximate positions of *SUS* genes of each species. Gene pairs with syntenic relationships are joined by red lines. The scale bar on the chromosome indicates chromosome length (Mb).

## Expression profile of pomegranate *SUS* family genes, assessed with RNA-seq and qPCR

In order to analyze the molecular functions of the *SUS* genes in pomegranate, we studied the transcript characteristics of the *PgSUS* genes using RNA-seq data downloaded from the NCBI SRA database (Fig. 5). For transcriptome analysis, a total of 300.88 Gb clean data with an average of 94.49% bases scoring Q30 were obtained from 42 RNA-seq libraries. The GC content of all samples ranged from 49.50 to 52.80%. It was found that more than 96% of the reads aligned with the pomegranate genome sequence, indicating a high sequencing quality and that the resulting data was reliable for subsequent analyses (Data S12).

*PgSUS* genes exhibited an obvious tissue specific expression pattern (Fig. 5A). The *PgSUS* family members of the SUS I and SUS II subgroups were predominantly expressed

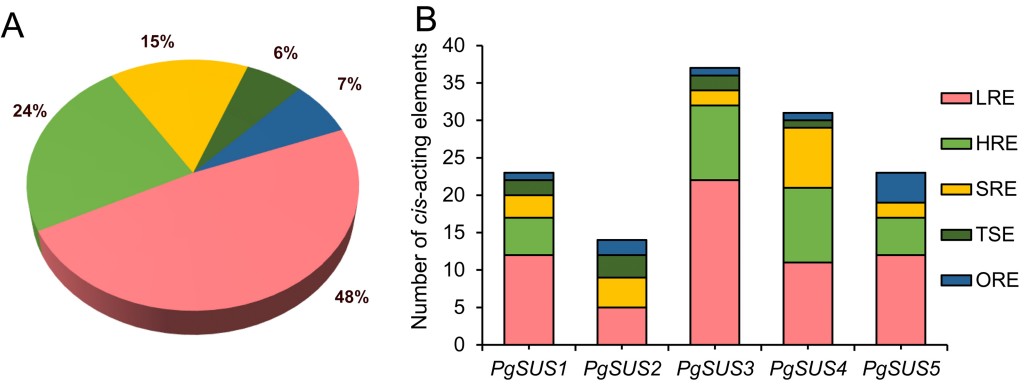

**Figure 4 Analysis of *cis*-acting element numbers in promoter region of five *PgSUS*.** *Cis*-acting elements of *PgSUS* genes were classified into five groups, including hormone responsive elements (HRE), tissue specific elements (TSE), light responsive elements (LRE), stress responsive elements (SRE), and other responsive elements (ORE). (A) Proportion of each functional group of *cis*-acting elements; (B) Number of *cis*-acting elements belonging to each functional group in individual *PgSUS* promoter sequences.

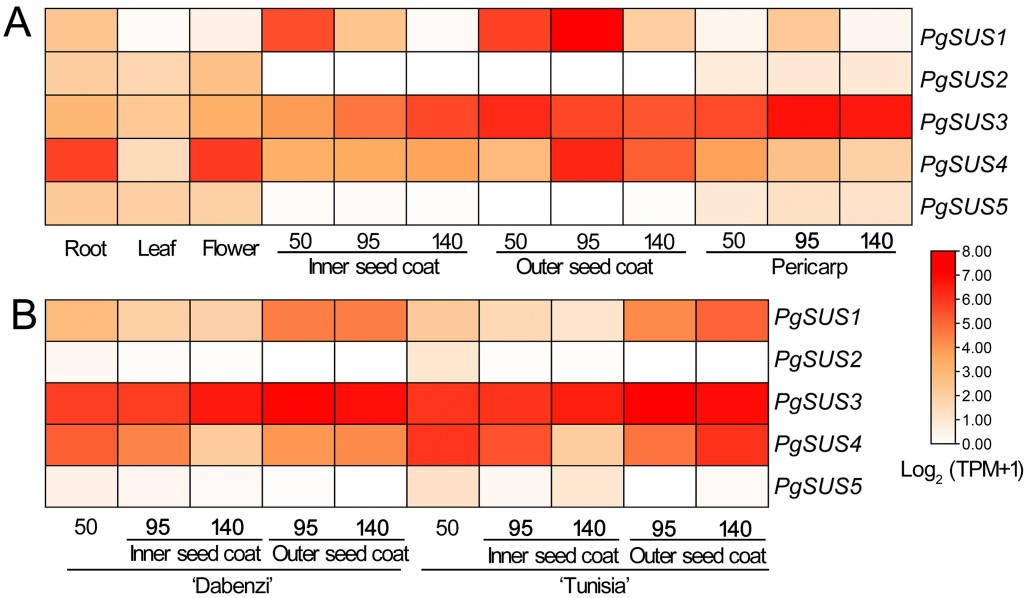

**Figure 5 Expression analysis of the *PgSUS* genes in different tissues of pomegranate.** (A) Expression profile of *PgSUS* genes in different organs or tissues of pomegranate, including root, leaf, flower, three stages of the pericarp, and the inner and outer seed coats (50, 95, and 140 DAF). (B) Expression profile of *PgSUS* genes at different developmental stages of the seed coats in cultivated pomegranate cultivars 'Dabenzi' and 'Tunisia'. The number represents the number of days after flowering (DAF). Expression levels are depicted in different colors based on Log$_2$-transformed TPM+1. White and red represent low and high expression levels, respectively.

in sink organs, particularly in fruit tissues. With the fruit development, *PgSUS1*, *PgSUS3*, and *PgSUS4* transcripts displayed different expression characteristics. *PgSUS1* was mainly expressed in the inner and outer seed coats, and reached its peak at 95 DAF in the outer

seed coat. The *PgSUS3* transcript was expressed at higher levels in the seed coat and pericarp (Fig. 5A). As the pericarp developed from 50 DAF to 95 DAF, *PgSUS3* expression gradually increased to the highest level, but slightly declined at fruit harvest (Fig. 5A). Interestingly, *PgSUS4* was strongly expressed in the sink organ, including the outer seed coat, root, and flower. *PgSUS4* showed a similar expression trend with *PgSUS1* as the outer seed coat developed from 50 DAF to 140 DAF. Its abundance rapidly increased on the 95 DAF (Fig. 5A). However, *PgSUS2* and *PgSUS5* of the SUS III subgroup were slightly expressed in the root, leaf, and flower, but was almost undetectable in fruits tissues (Fig. 5A). Furthermore, similar expression trends of *PgSUS* genes were also observed during the fruit development in the 'Dabenzi' and 'Tunisia' pomegranate cultivars (Fig. 5B).

QPCR was used to analyze the expression patterns of *PgSUS* genes. The relative expression level of each gene in different organs or tissues was standardized with their expression level in the leaf (Fig. 6). All five genes were up-regulated in root and flower compared with their expressions in the leaf. The relative expression level of *PgSUS4* increased more significantly than the other *PgSUSs* genes in root and flower. During fruit development, expressions of *PgSUS1* and *PgSUS4* rapidly increased with a tendency toward to a gradual decrease as the seed coat developed from 45 DAF to 150 DAF, which peaked at 75 DAF. These results suggest that isozymes encoded by these two genes may be involved in catalyzing key aspects of sucrose metabolism in the fruit seed coat during the early- and middle- developmental stages. *PgSUS3* showed stable expression levels when the fruit seed coat developed from 45 DAF to 115 DAF. Additionally, *PgSUS3* showed higher transcript levels in the pericarp than other genes, indicating that *PgSUS3* may play an important role in sucrose metabolism during the development of the pomegranate fruit pericarp. However, the transcripts levels of *PgSUS2* and *PgSUS5* were slightly or not-at-all expressed during fruit development. Our results show that three *SUS* genes (*PgSUS1*, *PgSUS3* and *PgSUS4*) may contribute to the sucrose metabolism and fruit development.

## DISCUSSION

Sucrose is synthesized in the leaf and transported to sink tissues, where it participates in growth and development, carbohydrate consumption, or the synthesis of major storage products in sink organs. In pomegranate, several key enzymes or genes play roles in sucrose metabolism in multiple sink organ or tissues, such as vegetative shoot apices, unpollinated ovaries, and seed (*Meletis et al., 2019*; *Poudel et al., 2020*). However, the molecular function of the *SUS* gene family as one of the key genes of sucrose metabolism in pomegranate remains unknown. Recently, more members of the *SUS* gene family have been identified and characterized from different plant species using comparative genome approaches and research advances in plant whole-genome sequencing, assembly, and annotation (*Stein & Granot, 2019*). The number of *SUS* family members differed among plant species, however, the *SUS* family typically contained four to seven genes (*Stein & Granot, 2019*). We identified at least five *SUS* genes in the pomegranate genome belonging to typical genes in the SUS family (Table 1). The average length of the SUS polypeptide chain was approximately 800 amino acids and the monomers weight was approximately 90 kDa

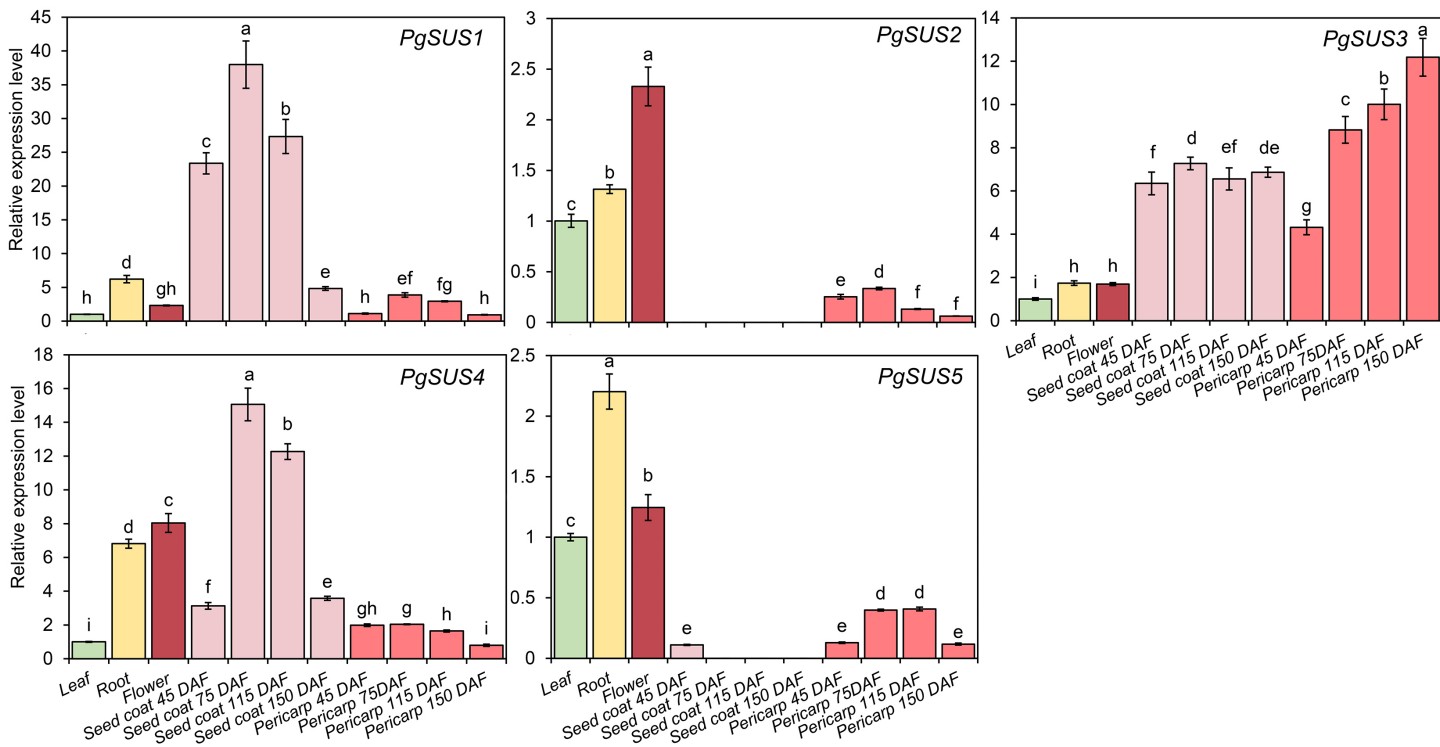

**Figure 6** **Expression pattern of five *SUS* genes assayed by qPCR.** *PgActin* served as the reference gene. Gene expression was normalized to the leaf expression level, which was assigned with a value of 1. Data represent the average of three independent replicates. Standard errors are shown as bars above columns. The different letters indicate significant differences at $p < 0.05$.

(*Stein & Granot, 2019*), such as in citrus CitSUS 1-6 (*Islam et al., 2014*), peach PpSUS1, 2, and 5 (*Zhang et al., 2015*), grape VvSS1-4 (*Zhu et al., 2017*), and pomegranate PgSUS1-4 (Table 1). The monomer weight of several other SUS isoforms was different with the members mentioned above. For example, the *Arabidopsis* AtSUS6 monomer weighs about 107 kDa (*Baud, Vaultier & Rochat, 2004*), and grape VvSS5 was estimated to be 102.7 kDa (*Zhu et al., 2017*). The weight of peach PpSUS6 (*Zhang et al., 2015*), Indian mustard BjSUS6.1, 6.2, 7.1, and 7.2 (*Koramutla et al., 2019*), and pomegranate PgSUS5 were estimated to be above100 kDa. Most of the pomegranate SUS proteins were predicted to be hydrophilic, with a low instability index, and contained acidic amino acids (Table 1), which was similar to the physical and chemical properties of other plant SUS proteins (*Islam et al., 2014*; *Tong et al., 2018*). Moreover, two putative Ser phosphorylation sites were observed in the N-terminal regions of all PgSUS proteins (Data S3), which may help determine the SUS subcellular localization and enzyme activity (*Stein & Granot, 2019*). Pomegranate SUS genes also shared a high sequence similarity of CDS and amino acid with 63 other SUS genes (Data S4). Therefore, the predicted molecular physicochemical characteristics of the five pomegranate SUS proteins were similar to be SUS proteins identified previously in other plant species.

The results of phylogenetic tree helped to predict the possible origin and relationships among different SUS isoforms. Although the SUS family genes shared high sequence

similarities (Data S4), phylogenetic result indicated that diversification occurred in this family. The SUS family has been historically classified into three major subfamilies in plants, namely SUS I, SUS II, and SUS III (*Xu et al., 2019*). The phylogenetic results of this study supported that the five *PgSUS* candidates were also categorized into distinct subgroups together with other SUS orthologs in *Arabidopsis* (*Baud, Vaultier & Rochat, 2004*), apple (*Tong et al., 2018*), and other species (*Stein & Granot, 2019*) (Fig. 1). SUS I was further classified into monocot- and dicot-specific subgroups (*Chen et al., 2012*; *Koramutla et al., 2019*; *Xu et al., 2019*). In pomegranate, *PgSUS1* and *PgSUS4* were clustered together with 17 other SUS genes of dicots into SUS I (Fig. 1), suggesting that a gene duplication event resulting in pomegranate *PgSUS1* and *PgSUS4* may have occurred after the split of monocots and dicots (*Chen et al., 2012*; *Koramutla et al., 2019*; *Xu et al., 2019*). Moreover, since *PgSUS1* and *PgSUS4* were grouped closely together and formed an independent pomegranate clade separate from *Arabidopsis*, pear, apple, peach, and other dicots genes. The independent gene duplication may have given rise to *PgSUS1* and *PgSUS4*, which may have occurred more recently after pomegranate's separation from *Arabidopsis* and *Rosaceae* species. The generation of the *PgSUS2* and *PgSUS5* genes, clustered together into SUS III, may have taken place before the separation of *Punicaceae/Arabidopsis*. We also observed the C-terminal extension in pomegranate SUS III subfamily genes (Data S5), implying that SUS III genes may derive from a common ancestor, which was consistent with previous studies (*Xu et al., 2019*). Additionally, *PgSUS3* and other members from both dicot and monocot species were grouped together into SUS II. These results support the view that SUS II and III subgroups are evolutionarily older than SUS I dicot subgroup (*Zhu et al., 2017*; *Chen et al., 2012*; *Koramutla et al., 2019*).

Intron and exon structures provide valuable information for the discovery of gene phylogenies (*Lecharny et al., 2003*). The intron loss event during ancient *SUS* genes evolution was proposed to be a common phenomenon, especially in the SUS I and SUS III gene subgroups (*Xu et al., 2019*). For instance, some introns may have been lost in *PgSUS1*, *PgSUS4*, *PgSUS2*, and *PgSUS5*. Intriguingly, the exon/intron structures of *PgSUS3* showed greater similarity to the putative ancestral genes of the SUS II subgroups (Data S6; Data S7), in which intron loss events occurred seldomly (*Xu et al., 2019*). These results support the hypothesis that the SUS II subgroup likely possessed relatively lower evolutionary rates (*Chen et al., 2012*; *Wang et al., 2015*; *Koramutla et al., 2019*; *Xu et al., 2019*). The additional exons in the 3′ end of *PgSUS2* and *PgSUS5* of SUS III were similar to the amino acid alignment (Data S5), leading to the complexity of intron/exon structure (Data S7). However, the function of the 3′ extension was unclear (*Xu et al., 2019*). Therefore, the evolutionary and functional effects of intron loss as well as the 3′ extension in the *SUS* genes requires additional research.

The motif composition and arrangement determinate the signature of the protein domain. SUS proteins showed conserved structural motifs among different plant species (*Zhang et al., 2015*; *Koramutla et al., 2019*). The motifs of five PgSUSs in this study shared extremely high similarities, suggesting that the pomegranate *SUS* genes were more conserved during evolution. Two common typical domains of SUS proteins were identified in several family members based on the similarity of nucleotide and peptide

chain sequences, conserved exons, and motif arrangements (*Zhang et al., 2015*; *Koramutla et al., 2019*), including five PgSUS proteins (Fig. 2C), which confirmed their authenticity in the pomegranate genome. The secondary and tertiary structure prediction of proteins provided the opportunity to obtain insights into understanding its biological functions (*Krissinel & Henrick, 2004*). Differences in the physicochemical characteristics of the protein sequences of five *PgSUS* genes resulted in their protein being folded into specific two- and three-dimensional structures (Data S9–Data S10). Among five pomegranate SUS proteins, the 2-D and 3-D of PgSUS1 and PgSUS4 were very similar, which was consistent with their close evolutionary relationship (Fig. 1) and implies that they may share similar functions. These results suggest that *PgSUS* family genes were highly conserved during evolution, despite the small differences found.

Whole-genome duplication (WGD), segmental duplication, and tandem duplication are the common gene duplication events in plants, which facilitated to gene family expansion and functional diversification (*Flagel & Wendel, 2009*). Although segmental or tandem duplication was suggested as the predominant pattern for the expansion of *SUS* family in pear (*Abdullah et al., 2018*), these types were not detected in *PgSUS* genes. This may explain the presence of relatively fewer *SUS* family members in pomegranate (Fig. 3; Table 1). In addition, four *PgSUSs* genes showed syntenic relationships with the genes of the other four species (Fig. 3), confirming their closer phylogenetic relationship, and their functional similarities.

In the gene promoter region, *cis*-acting elements may bind with specific transcription factors to modulate transcriptional levels, and respond to the stimulate signal (*Riechmann & Ratcliffe, 2000*). Light is an important environmental factor that may affect the storage or breakdown of sugars in roots, stems, and fruits in some biological metabolism, which then requires a series of enzymes, including sucrose synthase (*Girault et al., 2010*). In wheat, light illumination up-regulated the *SUS2* mRNA level, but decreased *SUS1* expression (*Maraña, García-Olmedo & Carbonero, 1990*). Compared with full-sun conditions, a higher expression of *CaSUS2* led to the improved hydrolytic activity of sucrose synthase in mature endosperm of coffee fruits under shade (*Geromel et al., 2008*). Here, the promoter prediction indicated that LREs occupied a larger proportion in the promoter region (Fig. 4A; Data S11), which was previously observed in Indian mustard and pear (*Koramutla et al., 2019*; *Abdullah et al., 2018*) These results indicate that light may be an important factor in the transcript regulation of *PgSUSs* genes. Moreover, research suggests that phytohormones may regulate the *SUS* expression level. In rice, *SUS* expression was induced by ABA during grain filling (*Tang et al., 2009*). In cotton, GAs promoted *GhSUSA1* expression, which resulted in the secondary cell wall deposition of fibers (*Bai et al., 2014*). The *SUS* gene was involved in the auxin-signaling pathway in tomato (*Goren et al., 2017*). Therefore, the presence of HREs predicted in the promoter region of pomegranate *SUS* genes implied their capacity to respond to phytohormones (Fig. 4; Data S11). *SUS* expression was also associated withstressors, such as low-oxygen, cold, heat, salinity, and drought (*Wang et al., 2015*; *Stein & Granot, 2019*; *Zhu et al., 2017*). SREs were found to be universally distributed in each *PgSUS* promoter, indicating that *PgSUS* genes may respond to abiotic stresses in pomegranate (Fig. 4B; Data S11). Therefore, predictive promoter analysis facilitates our

understanding of the multiple functions of *PgSUS* genes during pomegranate growth and development.

Several studies have shown that *SUS* genes exhibit tissue-specific and development-dependent expression profiles, primarily in the sink organs. *AtSUS2* was specifically induced in seeds (*Baud, Vaultier & Rochat, 2004*). The expression level of *ZmSUS3* gradually increased during the maize kernel maturation process (*Carlson et al., 2002*). The poplar *PtSUS* genes showed high transcript levels in roots, vegetative buds, and floral catkins (*An et al., 2014*). *VvSS1* expression in grape reached its peak at the start of young leaf development (*Zhu et al., 2017*). Likewise, the transcription and qPCR data presented in this study suggested the significant expression of some *PgSUS* genes (*PgSUS1*, *PgSUS3* and *PgSUS4*) in pomegranate sink organs (Figs. 5, 6). However, the expression levels of *PgSUS2* and *PgSUS5* were at low levels or undetected, indicating they might be redundant for pomegranate during the normal growth and development process. In edible fruits, the most important sink organ is fruit. *SUS* shows its close relationship with fruit development in several horticultural plants. For instance, *CitSUS1*, *CitSUS2* of the SUS I subgroup and *CitSUS6* of the SUS II subgroup were notably expressed in the juice sacs of citrus fruit (*Islam et al., 2014*). In peach, *PpSUS1* of the SUS I reached its highest levels during fruit maturation, while *PpSUS5* of SUS III was predominantly expressed in the early stages of fruit development (*Zhang et al., 2015*). *PbrSUS2* and *PbrSUS15* of SUS I were significantly up-regulated in pear fruit (*Lv et al., 2018*). *MdSUS1.1/1.2/1.4* of SUS I and *MdSUS2.1* of SUS II were mainly expressed in young and mature apple fruits, respectively (*Tong et al., 2018*). In this study, the significant expression of *PgSUS1*, *PgSUS4*, and *PgSUS3* were detected in different fruit tissues (Fig. 5, Fig. 6). The expression on *PgSUS1* in SUS I was quite high in early and mid-development stages of the fruit seed coat, which is the main edible part of the pomegranate. These results are consistent with the *MdSUS1.1* expression pattern, and confirms their evolutionary and syntenic relationships (Figs. 1, 3, 5 and 6). *PgSUS3* and *PgSUS4* were also highly expressed in the seed coat, with differential but partially overlapping expression patterns. Therefore, *PgSUS1* and *PgSUS4* of SUS I and *PgSUS3* of SUS II may play important regulatory roles in sucrose metabolism in the seed coat, as well as the quality of the fruit. These results also confirmed the molecular function of several members clustered into the SUS II subgroup that may overlap with the SUS I genes in specific plants (*Xu et al., 2019*). *PgSUS3* expression was notably increased in the pericarp, implying that *PgSUS3* could be closely related with the sucrose metabolism of the fruit pericarp *PgSUS4* was highly expressed both the root and flower, suggesting that *PgSUS4* may specially regulate sucrose metabolism in these sink organs with the exception of its functional redundancy in fruit development. These results imply that *SUS* genes in pomegranate may play crucial roles in pomegranate sucrose metabolism, particularly in fruit development.

## CONCLUSIONS

Our results show that the five sucrose synthase genes identified in the pomegranate genome, were clustered into three distinct subgroups. The structures of different *SUS* genes

in pomegranate were highly conserved during evolution and they might play different roles in sucrose metabolism and fruit development due to their partially overlapping but distinctly variable expression patterns. Our results further the understanding of the molecular basis of sucrose synthase genes in pomegranate. Future studies, including the analysis of gene overexpression and suppression, are needed to determine the specific functions of *PgSUS1*, *PgSUS3*, and *PgSUS4* in fruit sugar metabolism.

### Funding

This work was financially supported by the Natural Science Foundation of Anhui Province (No.1908085QC108), and the University Natural Science Research Project of Anhui Province (No.KJ2020A0043). The funders had no role in study design, data collection and analysis, decision to publish, or preparation of the manuscript.

### Grant Disclosures

The following grant information was disclosed by the authors:
The Natural Science Foundation of Anhui Province: No.1908085QC108.
University Natural Science Research Project of Anhui Province: No.KJ2020A0043.

### Competing Interests

The authors declare there are no competing interests.

### Author Contributions

- Longbo Liu performed the experiments, analyzed the data, prepared figures and/or tables, authored or reviewed drafts of the paper, and approved the final draft.
- Jie Zheng conceived and designed the experiments, authored or reviewed drafts of the paper, and approved the final draft.

### DNA Deposition

The following information was supplied regarding the deposition of DNA sequences:
The PgSUS gene sequences are available at GenBank: XM_031527544.1, XM_031535599.1, XM_031544401.1, XM_031516902.1, and XM_031551757.1.

### Data Availability

The raw data are available in the Supplementary Files.

### Supplemental Information

Supplemental information for this article can be found online at http://dx.doi.org/10.7717/peerj.12814#supplemental-information.

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
