# Peer review of "Identification and expression analysis of the sucrose synthase gene family in pomegranate (Punica granatum L.)"

_PeerJ, doi:10.7717/peerj.12814_

## Round 0.1 · original submission · Major Revisions

Please see the comments from reviewers.

Reviewer 1 ·

Basic reporting

Details about RNAseq should be presented. In materials and methods, it should be discussed about tissues, replications and sequencing analysis. In results sections, please present about DEGs and other details of transcriptome.

Experimental design

n materials and methods, it should be discussed about tissues, replications and sequencing analysis.

Validity of the findings

In results sections, please present about DEGs and other details of transcriptome.

Reviewer 2 ·

Basic reporting

There are some other papers that already dealt with different aspects of Sucrose metabolism in Pomegranate species (Meletis et al., 2018; Poudel et al., 2020). They should have been included in the discussion in order to provide a sufficient field context.

Experimental design

The study mostly relies on published data. The qPCR analyses presented in this study cannot be considered a “verification” of the data of Luo et al., 2020. Even if is the same var. and, possibly, the same plants, they were carried out in different experiments times.

Validity of the findings

The bioinformatic methodology used in this study is not up-to-date. Firstly, the TopHat read mapper has entered low maintenance since 2016 as it is now largely superseded by HISAT2 (https://ccb.jhu.edu/software/tophat/index.shtml). So the RNA-seq analysis should be entirely reconsidered. Moreover, I would suggest the authors not use FPKM (fragments per kilobase of exon model per million mapped reads) for transcript quantification since it should be used for comparing within-sample expression transcript expression levels. While TPM (Transcripts per Million) effectively normalizes for the differences in the composition of the transcripts in the denominator rather than simply dividing by the number of reads in the library. So that TPM is considered a more reliable parameter when comparison across samples is carried out.

Reviewer 3 ·

Basic reporting

In the study titled with “Identification and expression analysis of sucrose synthase gene family in pomegranate (Punica granatum L.)” by Longbo Liu et al. reported the identification of five sucrose synthase (SUS) genes from the pomegranate genome. With the phylogenic and motif analysis, they found that the PgSUS family were highly conserved during evolution. The author also predicted the protein structure and cis-regulatory elements of PgSUS1-5 genes and further investigated the spatial expression profiles of PgSUS family genes with public RNA-seq data and verified with Q-PCR.

The manuscript contains some interesting analysis; however, it is clear the author does not have a strong grasp of the English language and this makes it hard to follow. The author needs to improve the overall writing of manuscript with either professional English language editing service or benefit from someone of native English-speaking areas.

The overall structure of the manuscript conforms to journal standards and discipline norm except the discussion section. The literatures in introduction and other sections were well referenced & relevant with the correct format. Most figures were presented and showed in high quality and well labelled with raw data.

Experimental design

The experimental design including material preparation and bioinformatic software application in the manuscript is suitable to define questions which the author wants to demonstrate. The author described the methods with sufficient detail & information to replicate. At the same time, the author should present the methods in a more precise or accurate way which will make it more friendly to broader readers.

Validity of the findings

The most important finding in this manuscript is the identification of five PgSUS genes from the available genome pomegranate sequence. All the underlying data have been provided and statistically sound and controlled. Most conclusions are well stated, linked to the original research questions and supported by the results provided in the manuscript.

Additional comments

1)In the identification of PgSUS genes section, the author used six Arabidopsis SUS protein as the query sequences and obtained 19 candidates. After that, five candidates were excluded as they are carrying the sucrose-phosphatase domain. Could the author explain in detail in either results or method sections about how they filtered out the other 9 candidates? What were their criteria and definition of “redundant sequences”?

2) In my opinion, the figure 3 delivered very limited amount of information and should be placed in supplementary information or deleted from the main manuscript. Had the author also predicted the structure of SUS proteins from other species? If yes, are there any structure similarities between them? The author should provide this information if possible.

3) I would suggest the author to combine the figure 6 and 7 as both figures were used to demonstrate the spatial expression profile of five PgSUS gene. The heatmap data in figure 6 clearly indicates that certain PgSUS gene showed a tissue specific expression pattern. As the x-axis value is quite different in current figure 7 plot which make it impossible to compare the expression level of different PgSUS genes in the same tissue. Could the author re-organize or replot the data figure 7 so that it is clear to evaluate the five different PgSUS genes in the same organ?

4)I would suggest the author to rewrite the discussion section as it is very long and impossible to read. There are lots of detailed results information in this section and make is more like a supplement of results instead of discussion.



Minor points:
1)The last sentence in method sections of abstract is misleading, the author actually obtained the expression profiles of five PgSUS from previously published data. The author should clarify it.

2)Line 73-79, the description about SUS gene numbers in different species need further modification to make it more comprehensible, not confusing.

3)Line 178-179, The sentence “Each cDNA solution diluted 10 times was used as qPCR templates” is very confusing. The author should provide the exact amount of starting total RNA sample for cDNA synthesis.

4) The author should unify the description of quantitative real-time PCR as either qPCR or qRT-PCR.

5)Line 198-199, the sentence “Chromosomes 2, 4, and 6 contained only one gene, and chromosomes 8 contained two genes, respectively.” is improper and misleading. It could be change into “Among them, one single SUS gene was originated from Chr2, 4 and 6 respectively, while the rest two was located in Chr8”.

---

## Round 0.2 · accepted · Accept

Based on recommendations from the reviewers, this manuscript can be accepted.

Reviewer 2 ·

Basic reporting

The new version of the manuscript is in line with the provided suggestions.

Experimental design

The new experimental design description is clear and scientifically solid.

Validity of the findings

The findings presented in this study have a good relevance for sucrose metabolism investigation in pomegranate research being a spillover for the improvement of this crop through breeding programs.

Reviewer 3 ·

Basic reporting

The authors figured out most of the issues I concerned and made detailed responses to my comments. In view of the improved manuscript with much better quality, I suggest the publication of this work in this journal.

Experimental design

No comment.

Validity of the findings

No comment.

Additional comments

No comment.